# State of the Art and Challenges in Epilepsy—A Narrative Review

**DOI:** 10.3390/jpm13040623

**Published:** 2023-04-01

**Authors:** Aida Mihaela Manole, Carmen Adella Sirbu, Mihaela Raluca Mititelu, Octavian Vasiliu, Lorenzo Lorusso, Octavian Mihai Sirbu, Florentina Ionita Radu

**Affiliations:** 1Department of Neurology, ‘Dr. Carol Davila’ Central Military Emergency University Hospital, 010242 Bucharest, Romania; 2Clinical Neurosciences Department, University of Medicine and Pharmacy “Carol Davila” Bucharest, 050474 Bucharest, Romania; 3Centre for Cognitive Research in Neuropsychiatric Pathology (Neuropsy-Cog), Department of Neurology, Faculty of Medicine, “Victor Babeș” University of Medicine and Pharmacy, 300041 Timișoara, Romania; 4Nuclear Medicine Department, ‘Dr. Carol Davila’ Central Military Emergency University Hospital, 010242 Bucharest, Romania; 5Department No.8, University of Medicine and Pharmacy “Carol Davila” Bucharest, 050474 Bucharest, Romania; 6Department of Psychiatry, ‘Dr. Carol Davila’ Central Military Emergency University Hospital, 010242, Bucharest, Romania; 7Neurology Unit—Neuroscience Dept. A.S.S.T.Lecco, Merate Hospital, 23807 Merate, Italy; 8Department of Neurosurgery, ‘Dr. Carol Davila’ Central Military Emergency University Hospital, 010242 Bucharest, Romania; 9Department of Gastroenterology, ‘Dr. Carol Davila’ Central Military Emergency University Hospital, 010825 Bucharest, Romania; 10Department of Gastroenterology, “Carol Davila” University of Medicine and Pharmacy, 020021 Bucharest, Romania

**Keywords:** seizure, seizure-free, quality of life, epilepsy, education, drug-resistant, medically intractable, pharmacoresistant, surgery, neurosurgery

## Abstract

Epilepsy is a common condition worldwide, with approximately 50 million people suffering from it. A single seizure does not mean epilepsy; almost 10% of the population can have a seizure during their lifetime. In particular, there are many other central nervous system disorders other than epilepsy in which seizures occur, either transiently or as a comorbid condition. The impact of seizures and epilepsy is, therefore, widespread and easily underestimated. It is estimated that about 70% of patients with epilepsy could be seizure-free if correctly diagnosed and treated. However, for patients with epilepsy, quality of life is influenced not only by seizure control but also by antiepileptic drug-adverse reactions, access to education, mood, employment, and transportation.

## 1. Introduction

Epilepsy is a heterogeneous disorder characterized by epileptic syndromes, diverse etiologies, and variable prognosis. Epileptic seizures are quite common, affecting between 8 and 10% of the population throughout their lifetime and accounting for 1–2% of presentations to an emergency room, and about a quarter of these will be first seizure with a different type (Table 1) [1,2].

## 2. Definition and Terminology

Epileptic seizure has been defined by the International League Against Epilepsy (ILAE) as the transient occurrence of signs and/or symptoms as a result of abnormal, excessive, or synchronous neuronal brain activity [3]. The definition of epilepsy has undergone some changes, so since 2014, the ILAE has proposed the inclusion of one of the following [4]:At least two unprovoked (or reflex) seizures occurring more than 24 h apart;An unprovoked (or reflex) seizure and the likelihood of subsequent seizures similar to the overall risk of recurrence (at least 60%—accompanied by clinical, electrical-electroencephalogram (EEG), or neuroimaging changes) after two unprovoked seizures occurring within the next 10 years;Diagnosis of epilepsy syndrome [5].

## 3. Etiology

While some causes of seizures (Table 2) can affect children of any age, others have a predilection for certain age groups. In newborns, for example, most of them are symptoms of an identifiable etiology, such as neonatal encephalopathy, a metabolic disorder, or a systemic infection of the central nervous system. In older infants and young children, febrile seizures are the most common, age-dependent cause [5]. A structural etiology is determined when an abnormality is seen on neuroimaging and when the signs and symptoms of seizures, in combination with electroencephalogram (EEG) data, suggest this abnormality is the probable cause of the seizures. If the clinical and EEG data are discordant with the localization of the visible anatomical abnormality, then the imaging modification is not relevant to the patient’s epilepsy. The structural difference may be genetic, acquired, or both.

## 4. Clinical Aspects

Seizures in younger children are significantly different from those in older children and adults (Table 3). Those older than 6 years tend to have seizures similar to adults, while younger children have fewer complex behaviors, especially focal seizures with impaired consciousness [6].

Focal epilepsy can develop at any time of life, and the etiology varies according to age, as does the semiology vary according to location as well:

**Temporal lobe epilepsy** is one of the most typical regions. Many patients present with aura, which may include „deja vu,” „jamais vu,” and „butterflies in the stomach”—epigastric aura, fear, dyscognitive phenomena, or olfactory symptoms [12]. The aura is usually followed by impaired/maintenance of consciousness but with the appearance of oromandibular or brachial automatisms or even dystonic postures. Symptomatology associated with the nondominant lobe may include nausea and vomiting. The seizure lasts between 60 and 90 s and is followed by a state of confusion which predominates in patients with dominant-lobe damage [13]. If the focus is in the lateral area of the temporal lobe, perisylvian, the eloquent areas, Wernicke’s area, or primary and secondary auditory cortex are affected, leading to an aphasia-type language disorder or auditory aura. Localization at the temporo-occipital junction may associate with vertigo symptoms or visual aura. The aura may progress to a capped gaze followed rapidly by bilateralization [14].

**Frontal lobe epilepsy** is the second-most-frequent location. Compared to the previous one, seizures are predominantly motor, of shorter duration, and have a nocturnal onset. The postictal period is characterized by a rapid return of awareness and may occasionally be associated with a motor deficit. Regarding semiology, it is divided into the primary motor area, somatomotor area, orbitofrontal, dorsolateral, and opercular regions. Most of them are associated with predominantly motor phenomena, but those in the posterior, opercular zone can easily be mistaken for temporal lobe epilepsies through gaze capping, automatism, and head and eyeball versa. Asymmetric tonic seizures, which are described as tonic flexion of one arm and extension of the other, with or without tonic involvement of the lower limbs, are associated with activation of the Brodmann area [15,16].

**Parietal and occipital lobe epilepsy**. In this case, this type of localization is much rarer than the others. The occipital region is associated with elementary visual hallucinations, unlike the temporal lobe, where complex visual hallucinations occur [17]. Parietal lobe epilepsy may present with a somatosensory aura, which may be unilateral or bilateral. Both types can spread rapidly and may mimic/associate semiological elements [18].

## 5. Diagnostic of Epilepsy

First of all, the diagnosis of epilepsy is a clinical one, and the other additional tests are supportive. For an accurate diagnostic, there are some paraclinical tests illustrated in Table 4. The first thing to consider in correctly diagnosing epilepsy is to determine whether a paroxysmal clinical event is actually an epileptic seizure or another pathology. For the differential diagnosis, we must consider all the causes of episodes of altered consciousness, altered mental status, motor or sensory manifestations, and seizures, which are common in other epilepsies. For epileptologists, it is often easy to recognize different forms of epilepsy when they are able to obtain a clear history of events. However, at the same time, even the most experienced epileptologists have great difficulty in reaching an unequivocal diagnosis for various reasons, such as atypical seizure presentations, inadequate or incomplete historical data, or overlapping symptomatic manifestations.

Thus, the misdiagnosis of epilepsy is a huge medical problem with a significant impact on the patient. Many disorders can mimic epileptic seizures (Table 5), but this is also possible the other way around; some seizures can mimic symptoms of other diseases. This can happen in patients with non-epileptic disorders who may be misdiagnosed and automatically mismedicated. Also, patients with epileptic seizures may be misdiagnosed as migraine, encephalitis, or other paroxysmal non-epileptic events and are likely to be mismanaged with treatments that cannot help them and also deprived of specific therapies [26].

## 6. Evolution, Complication, and Prognostic

In general, prognosis refers to the likelihood of not having a seizure after starting the treatment. Most people have a good prognosis for complete seizure control and eventual discontinuation of antiepileptic medication, but epilepsy syndromes have different treatment responses and outcomes. Prognostic factors include etiology, EEG pathology changes, seizure type and number, and therapeutic response. Early feedback to treatment is an important long-term positive predictor, while a history of a large number of seizures at diagnosis, intellectual disability, and symptomatic etiology are negative predictors. Patients with epilepsy have a higher mortality risk than the general population [29].

Attention is paid to reducing seizure frequency, and severity, and to treatment management. However, quality of life is also determined by other factors: impact of diagnosis, adverse effects of medication, driving, education, employment, and independence. 

Patients with this pathology have a higher risk of psychiatric diseases, including depression, anxiety, and attention-deficit disorders. Despite the implications of life quality, depression in epilepsy is under-detected and underdiagnosed. In this category, we can add cognitive impairment, a common problem. It depends on several factors: seizure frequency, type of treatment, early onset, interictal discharges, educational level, and polypharmacy. It can manifest differently depending on the area or areas affected, such as memory disorder, executive dysfunction, slowed psychomotor speed, sustained attention-deficit disorder, visual-spatial disorder, and anomic aphasia.

Another complication can be an increased risk of injury, which is attributable to seizures due to falls. Additional risk factors are epilepsy not under therapeutic control, seizures with altered consciousness, and therapeutic neglect. Sequelae can lead to shoulder dislocation, vertebral fractures, and in more severe cases, subdural hematoma [30].

The most often cause of death straight associated with epilepsy is SUDEP, sudden unexpected death in epilepsy, in which autopsy reveals no other anatomical or toxicological explanation. A couple of risk factors have been identified: the presence of generalized tonic–clonic seizures, >3 generalized tonic–clonic seizures per year, and uncontrolled, pharmacoresistant epilepsy. The majority of patients, around 70%-80%, are well-controlled therapeutically. Pharmacoresistant epilepsy is, according to the ILAE, failure of two tolerated antiepileptic drugs, appropriately chosen and used, either as monotherapy or in combination [31]. One of the neurological emergencies is *status epilepticus*, which is associated with a higher probability of mortality and morbidity. It is not a single entity; it has a number of different aspects, forms, and a wide range of etiologies. Currently, the accepted definition is either a seizure lasting at least 5 min or two or more seizures with incomplete recovery of consciousness. In contrast, non-convulsive *status epilepticus* is defined as *status epilepticus* without prominent motor symptoms lasting more than 10 min [32]. Refractory *status epilepticus* refers to the clinical and electroencephalographic persistence of seizures after initiation of line 1 benzodiazepine and line 2 treatment. *Super refractory status epilepticus* is defined by a prolonged frequency of seizures 24 h after initiation of anesthetic treatment. Suportting the factors described above, some studies have shown an increased incidence of seizures directly proportional to environmental factors such as ambient temperature, methane (CH4), and nitric oxide (NO). Moreover, the average daily number of hospital visits was significantly high in January and February than in other months of the year. Some seizures may be caused by an underlying etiology, such as cerebral hemorrhages, and the risk of their occurrence depends on variations in temperature and atmospheric humidity [33].

## 7. Healthcare of people with Epilepsy

There are different types of epilepsy involving the frequency and severity of seizures. By their nature, they are often unpredictable and sporadic. The purpose is to keep the equilibrium between restrictions planned to keep health and safety during the ictal period with sustaining a full and active lifestyle during seizure freedom. Quality of life is determined not only by seizure control but also by side effects of medication, relationships, school education, employment, and transport. Well-being not only includes illness-therapeutic control but also mental state, lifestyle elements such as employment, and resources. A secondary reaction is the major impact as a financial cost to society. In the USA, the total spending on this disease reached 12.5 billion dollars, of which 85% is the not directly related expenses. For these people, these costs are also countable in days lost from work and school, unemployment, social isolation, poor social life, and minimal emotional and financial support [34]. 

One of the major issues is the high incidence of psychiatric disorders. Meta-analyses of population-based studies indicate a cumulative lifetime prevalence of 23% for depression and 20% for anxiety. In addition, they have been shown to be twice as likely to have suicidal thoughts and up to 3 times more likely to die by suicide compared to the general population [35,36]. Unfortunately, depression can occur at any time of life or age and can precede the onset of seizures. Furthermore, the severity of depression is directly proportional to an increased likelihood of uncontrolled seizures and has been correlated with lower adherence to antiepileptic drugs [37]. Even though the implications are potentially serious in terms of quality of life, depression in epilepsy is underdiagnosed. One reason for this may be the difficulty in differentiating between its symptoms: fatigue, insomnia, and cognitive decline, from the side effects of antiepileptic therapy. In this context, research has shown that in terms of cognitive impairment, older-generation antiepileptics (carbamazepine, valproate) are clearly inferior to newer ones (lamotrigine and levetiracetam). Topiramate has the greatest negative effects on cognition. In order to prevent or keep these disorders under control, health professionals can perform neuropsychological testing. This can pinpoint areas of deficiency for the patient, and then we can develop strategies to overcome them (i.e., treatment optimization, memory exercises: meditation, crosswords, puzzles, Sudoku, chess). Another problem is regarding the family members, who may also attribute these signs to a “normal” response to seizure-induced limitations and, therefore, will not inquire about any investigation. Recognition can be improved by the use of screening tools for depression that have been validated in these cases [38]. The choice of antiepileptic therapy can also have an impact on mental status, for better or worse. One expert recommendation in this regard includes the use of an antiepileptic with mood-stabilizing properties, such as lamotrigine or valproate, as well as avoiding drugs with emotional side-effect-increasing properties. Lamotrigine has been shown in two randomized, double-blind trials to improve emotional status after 7 to 8 weeks of treatment [39,40]. Another issue with emotional impact is driving. In a survey of people with epilepsy, driving was rated as the main concern that impacts the quality of life. The ability to drive can influence work performance, preserve relationships, and live independently. It is not difficult to conceive that a seizure with loss of consciousness while driving could have catastrophic effects [41]. Seizures with impaired consciousness can lead to burns, drowning, and motor vehicle accidents. Most related incidents are of low to moderate severity and generally include lacerations, fractures, dental injuries, contusions, and burns. Serious injuries such as subdural hematomas or death by drowning occur, but rarely [42].

One of the most important strategies to reduce the risk of injury is to improve seizure control. Risk strategies for prevention must be personalized to the direct factors closely associated with epilepsy, such as seizure characteristics, recurrence, timing (sleep versus wake), and other triggers. It is also important to consider data appropriate to the patient profile, such as age, cultural and social norms of the patient, and family. Too many measures for prevention may determine the lack of involvement in healthy habits, which reduces their social life and may inhibit the patient from achieving autonomy. A few strategies have been selected to reduce the risk of seizure-related injury: Wear a helmet when cycling or horseback riding;No unsupervised swimming;It is advisable to use the shower, not the bathtub;Water temperature control of the hot water heater to reduce the risk of scalding;Use a microwave oven, not a stove;Avoid locking the toilet or bedroom door;The height of the bed should be small in order to avoid possible post-crash injuries;It is preferable to use an epilepsy safety pillow (made to reduce the risk of suffocation if sitting face down);Be careful of high stairs;Beware of driving rules for patients with epilepsy;Administration of medication according to the scheme-continuum.

An important problem for this kind of patient is the shift from childhood to adulthood. Transition involves multiple medical resources to provide for the young adult’s psychosocial, educational, and therapeutic needs. At a minimum, this involves the coordination between health professionals to ensure proper continuity of treatment management and ensuring the transfer of the necessary information about the patient. To the extent possible, they should achieve some skills in different types of areas of self-care so that they are prepared to increase their empowerment regarding their healthcare needs. In children with inherited metabolic disease, the elementary healthcare provider is commonly either a pediatric neurologist or a metabolic pediatrician. While in the changeover from pediatric to adult neurologist, the biggest issue may be exaggerated patient and/or family anxiety, which is entirely resolved with proper coordination, the transition between metabolic medicine specialists is usually difficult due to a deficit of adult specialists in this domain [43]. EuroNASH: European Audit of Seizure management in Hospitals and the European Study of Burden and Care in Epilepsy (ESBACE) will provide guidance for better management and use of investigations in people with seizures [44]. These things combined draw our attention to the needs of these patients and the care we need to take in managing both the underlying disease and adjacent ones, which may or may not be secondary to treatment.

## 8. Treatment of Epilepsy—Principles of the Current Status

The major determinants in the selection of antiepileptic therapy are the types of seizure and epilepsy. Classification of seizures into focal and generalized help us in choosing the appropriate treatment. However, in children, it has specific age-dependent characteristics, with seizure types and syndromes involving therapies rarely used in adults. There are some syndromes, such as Lennox–Gastaut, where seizure frequency is very high and requires polytherapy, but there are also childhood epilepsies with benign character, centrotemporal and occipital spike epilepsy, where long-term medication is not required.

The decision to initiate an antiepileptic lean on whether the diagnosis of epilepsy is confirmed. Aspects that help raise the risk of recurrence implicate the presence of an abnormal clinical examination, imaging abnormalities, nocturnal seizures, and abnormal EEG.

Two types of seizures in children, epileptic spasms and typical absence seizures, have unique treatment options. Factors that help in choosing the right antiepileptic depend on:Epilepsy characteristics: seizure type, seizure frequency, specific epilepsy syndrome;Patient characteristics: gender, age, comorbidities, pregnancy, allergies, current and previous medication;Drug characteristics: efficacy, adverse reactions, drug interactions, half-life, titration, risk of teratogenicity, interaction with oral contraceptives, liquid/solid form;Socio-economic characteristics: cost, availability, personal choice.

Results of the meta-analysis demonstrated that for focal seizures, lamotrigine and levetiracetam were considerably superior to carbamazepine, which was better than phenytoin and phenobarbital. Regarding generalized seizures, valproic acid was superior to topiramate, carbamazepine, phenytoin, and phenobarbital. In newly diagnosed focal epilepsies, carbamazepine and lamotrigine were superior, and levetiracetam and perampanel had some advantages in treatment-refractory epilepsies [45].

## 9. Drug-Resistant Epilepsy

Epilepsy is considered drug resistant if at least two appropriately chosen and used antiseizure medications have failed to control seizures [46].

Various anticonvulsant drugs are available for the treatment of focal epilepsy with seizures refractory to a first or alternative monotherapy. Anticonvulsant drugs that are currently used in clinical practice as adjunctive treatments include lamotrigine, oxcarbazepine, levetiracetam, pregabalin, clobazam, zonisamide, eslicarbazepine acetate, brivaracetam, gabapentin, lacosamide, topiramate, valproate, vigabatrin, and perampanel [47].

In cases with idiopathic generalized epilepsy, approximately 35% of them will require adjuvant therapy. In some studies, perampanel has been demonstrated to reduce seizure frequency by 76.5%. Other antiepileptics such as Lamotrigine, Levetiracetam, and Topiramate have shown efficacy in first generalized tonic–clonic seizures not controlled by a first or additional monotherapy [48].

There are situations where monotherapy cannot succeed, and the addition of a second antiepileptic is necessary. The choice of an adjuvant is difficult because issues of efficacy, tolerability, pharmacokinetic properties, drug interactions, and frequency of administration must be taken into consideration. Therefore, in order to have an ideal and rational polytherapy, some criteria should be met: synergistic effects (their mixed effectiveness should be higher than the sum of the efficacy of each individual anticonvulsant drug) and infra-additive toxicity (their summed toxicity should be lower than the individual one). Preferred combinations are those that have different mechanisms of action or that have multiple mechanisms of action [49]. For generalized epilepsies, the combination with the maximal evidence for synergistic potency is valproic acid and lamotrigine. Patients who have not reacted to the top-tolerated dose of lamotrigine or valproic acid may have seizure control by combining them. In some patients with Dravet syndrome, Lennox–Gastaut syndrome, cannabidiol can be used, an enzyme-inhibiting drug that can boost serum concentrations of the active metabolite of clobazam [50].

Other options for patients with pharmacoresistant epilepsy include surgery, the use of neurostimulation (vagus nerve stimulation (VNS), responsive neurostimulation (RNS), deep-brain stimulation (DBS)), and modern minimally invasive techniques, laser interstitial thermal therapy (LITT), and stereotactic radiosurgery.

Candidates with focal epilepsy for resective surgery can be divided into:Mesial temporal lobe epilepsy or neocortical epilepsy;Lesional epilepsy due to focal structural pathology (low-grade glioma, cavernous malformation);Nonlesional focal epilepsy.

Patients with drug-resistant focal epilepsy require pre-surgical evaluation to properly determine and define the epileptogenic area to be removed, subsequently to have a chance to be seizure-free. For this purpose, a history is taken with the semiology of epileptic seizures, frequency, and duration in order to better understand their location and epilepsy subtype. An important marker is video-EEG monitoring to observe interictal, ictal changes, and correspondence between symptoms and location of seizure onset. In some situations, antiepileptic treatment is reduced to increase the possibility of seizure recording and make sure that the patient is having only one type of seizure [51,52].

High-resolution brain imaging with epilepsy protocol is necessary to detect potential abnormal structures that may be the cause of epileptic seizures. Neuropsychological testing is necessary to detect any pre-surgical deficits that may be correlated with the seizure-onset area and to predict possible postoperative deficits. Positron emission tomography (PET) has proven to be useful, especially in cases with negative MRI; there may be areas of hypometabolism that can confirm the epileptogenic site [53]. Once these investigations are completed, and there is a reliable correlation between the symptoms and their outcome, surgery can be performed. Otherwise, when in doubt, investigations should continue with intracranial EEG monitoring. Intracranial EEG is also necessary when the seizure-onset zone is close to the eloquent cortex.

Neurostimulation therapy may help patients with drug-resistant epilepsy who present contraindications for epilepsy surgery or have epileptogenic regions close to the eloquent areas.

In general, a significant amount of epilepsy exploration endeavors have been centered on the development of treatments and surgical interventions, but few clinical trials have evaluated medical services, and etiologies, and there has been less evaluation of the process of care and affiliated outcomes and costs.

It is well known that air pollution is also an important negative factor in exacerbating neurological pathologies, epilepsy being no exception. That is why we need to create new strategies and mechanisms to reduce its negative effects on the nervous system and mental health [54,55].

## 10. Conclusions

Informing patients about all aspects of this pathology, such as triggers of seizures, adverse reactions of treatment, and possible risks to which they are susceptible, can be considered an important step in the management and, implicitly, in easier integration into society. When doing so, it is important to consider the wider population, as those without specialist care are likely to have poorer outcomes, which could be improved if outpatient care were more accessible. These things combined draw our attention to the needs of these patients and the care we need to take in managing both the underlying disease and adjacent ones, which may or may not be secondary to treatment.

## Figures and Tables

**Table 1 jpm-13-00623-t001:** The expanded version of the 2017 ILAE seizure-type classification.

Focal Onset		Generalized Onset	Unknown Onset
Aware	Impaired Awareness		
**Motor Onset** Automatisms Atonic Clonic Epileptic spasms Hyperkinetic Myoclonic Tonic **Nonmotor onset** Autonomic Behavior arrest Cognitive Emotional Sensory		**Motor** Tonic-clonic Clonic Tonic Myoclonic Myoclonic-tonic-clonic Myoclonic-atonic Atonic Epileptic Spasms **Nonmotor (absence)** Typical Atypical Myoclonic Eylied myoclonia	**Motor** Tonic-clonic Epileptic spasms **Nonmotor** Behavior arrest
Focal to bilateral tonic-clonic			Unclassified

Caption: ILAE = International League Against Epilepsy.

**Table 2 jpm-13-00623-t002:** The causes of seizures and epilepsy.

Genetic	Structural	Metabolic	Immune	Infectious
o Childhood absence epilepsy o Juvenile absence epilepsy, and juvenile myoclonic epilepsy o Dravet syndrome	o Stroke o Tumor o Neurodegenerative diseases, Alzheimer o Vascular malformation o Head trauma o Neural development lesions, cortical dysplasia, hippocampal sclerosis	o Mitochondrial disease o Hyperthermia o Hyperglicemya o Hypocalcemia o Hyponatremia o Hepatic encefalopahty o Uremic encefalopathy	o Rasmussen encephalitis o Anti-NMDA receptor encephalitis o Autoimunne encephalitis o Anti–leucine-rich, glioma inactivated 1 (anti-LGI1) encephalitis	o Neurocysticercosis Human immunodeficiency virus (HIV) o Cytomegalovirus o cerebral toxoplasmosis o Tuberculosis

Caption: categories of etiology according to the International League Against Epilepsy (ILAE).

**Table 3 jpm-13-00623-t003:** The most common syndromes in children.

**West Syndrome**	- Occurs between 3 and 12 months. - Consists of the triad: epileptic spasms, hypsarrhythmia, and psychomotor retardation. - It may also be a precursor to Lennox–Gastaut syndrome [7].
**Dravet Syndrome**	- Onset in the first 18 months of life, which may be with focal seizures, some with secondary bilateralization, usually triggered by fever or hyperthermia. - Absence-type, myoclonic, focal, and reflex seizures may also occur in preschoolers on photostimulation or hyperventilation. - Initially, intellectual capacity is not affected, but over time, as the number of seizures increases, cognitive impairment can be severe [8].
**Febrile seizure**	- can be simple or complex - Simple febrile seizures are usually generalized, last less than 15 min, and do not recur over a 24-h period. - Complex seizures also occur after the age of 6 years, accompanied by fever or not, and may be the basis of Dravet syndrome, myoclonic–atonic epilepsy, and hippocampal sclerosis; they are focal, prolonged, and may recur over 24 h.
**Benign rolandic epilepsy**	- Is a type of self-limiting epilepsy, presenting with focal seizures with the clonic or tonic activity of a part of the face or tongue and oral paresthesia. - Focal seizures are more frequent in young children, and progression to bilateralization is frequent during sleep and may be associated with a post-critical deficit, Todd’s palsy. - Psychomotor development in this situation is normal [9].
**Typical absence seizure**	- The onset is sudden and lasts 20–30 s and can occur up to 50 times a day, and can be complicated in 40% of cases by generalized tonic–clonic seizures. - It may be associated with palpebral myoclonus and may be triggered by hyperventilation [8].
**Lennaux– Gastaut syndrome**	- Includes the triad: generalized, tonic, atonic, myoclonic, and atypical absence seizures, peak-wave complex EEG interictal changes, and psychomotor retardation. - Nocturnal tonic events are characteristic of this syndrome but are difficult to recognize without video-electroencephalogram monitoring. - Negative prognostic factors are psychomotor retardation preceding seizures, history of epileptic spasms, the onset of symptoms before the age of 3 years, increased seizure frequency, and recurrence of non-convulsive status epilepticus [10].
**Juvenile Myoclonic Epilepsy**	- Is the most common form of genetic epilepsy. - Patients usually present with a generalized tonic–clonic seizure, often caused by sleep deprivation, alcohol ingestion, stress, or intermittent light stimulation, although on a thorough history, most patients have had morning myoclonus [11].

**Table 4 jpm-13-00623-t004:** Paraclinical examinations in epilepsy.

Test	Result
**Electroencephalogram (EEG)** EEG helps us to complete the diagnosis of epilepsy, choose appropriate therapy, monitor response to treatment, and determine candidates for antiepileptic drug withdrawal and surgical localization.	Focal spikes or sharp waves with associated slowing of the electrical activity in the area of the spikes.
**Video-electroencephalogram (EEG) long-term monitoring (LTM)** Video-EEG recording is useful and indicated in patients with suspected psychogenic seizures, for epilepsy classification, and especially in those with pharmacoresistant focal epilepsy, possible candidates for epilepsy surgery. It is also helpful in intensive care units in the evaluation of encephalopathies and non-convulsive status epilepticus.	Capturing seizure activity simultaneously on video recording and EEG; increased EEG sampling may reveal evidence of interictal abnormalities (spikes and sharp waves), which may make the diagnosis of focal seizures more likely.
**CT head** Usually ordered in emergencies in patients presenting with a first seizure episode (37). It is useful for identifying acute causes of seizures but is less sensitive to smaller abnormalities often seen on MRI.	Structural lesions
**MRI brain** Neuroimaging helps us to identify the underlying etiology of focal or generalized seizures and the location of the epileptogenic area and to determine the surgical location in focal pharmacoresistant epilepsies [19]. The optimal MRI technique for patients with focal seizures is 3 Tesla studies with coronal, axial, and sagittal T1, T2, and FLAIR sections [20]. The epilepsy protocol should also include a 3D T1 with a volumetric acquisition, which allows better assessment of cortical dysplasia or discrete focal lesions [21]. FLAIR has a 97% accuracy in detecting abnormalities, especially temporal sclerosis [22,23].	Structural injuries (mesial temporal sclerosis, neoplastic lesions, vascular malformations, and developmental lesions).
**PET scan** Part of the surgical evaluation of treatment drug-resistant focal epilepsy.	Ictal-hypermetabolic; interictal-hypometabolic
**Laboratory test**	Extreme hypoglycemia or hyperglycemia can cause provoked focal seizures.
- Blood glucose - Toxicology screen - FBC - Electrolyte panel - Lumbar puncture is indicated when CNS infection or an immune etiology is suspected.	
**Genetic testing** Genetic testing is increasingly available for a number of inherited syndromes but has variable clinical utility depending on the clinical and genetic heterogeneity of a syndrome [24,25].	

**Table 5 jpm-13-00623-t005:** Epilepsy mimics.

Condition	Differentiating Signs/Symptoms
**Syncope**	Vasovagal syncope usually has prodromal sensations of dizziness, nausea, and diaphoresis, often caused by the change of position, physical exertion, Valsalva maneuvers, and strong emotional impact. The loss of consciousness is short-lived; if it persists, convulsive movements may occur, and confusion may arise between the diagnosis of syncope and epileptic seizure. Post-critical confusion and urinary incontinence are rare in this situation. Cardiogenic syncope is usually the result of bradyarrhythmia or tachyarrhythmia disorders [27].
**Cerebrovascular disease**	The clinical features depend on the duration of the ischemia, the territory of the vasculature affected, and the anatomical location. Transient ischemic attacks typically last from a few minutes to an hour, usually. Cerebrovascular disease is usually associated with negative signs such as muscle weakness, aphasia, and decreased visual acuity. Epileptic seizures are associated with positive signs in the ictal period but post ictally may have negative signs, confusing with stroke, in which situations-imaging and video-EEG clarify.
**Migraine**	May have clinical similarities similar to an epileptic seizure, such as visual, sensory, or dyscognitive phenomena. However, the mechanism is different, and it is quite rare for an epileptic seizure to follow a migraine.
**Movement disorders**	Dyskinesias, paroxysmal dystonia, or tremor can mimic a focal epileptic seizure. Normal EEG during these movements and carefully studied semiology differentiate the two pathologies.
**Sleep disorders**	Another easily confusing category is REM sleep disorders, parasomnias. Non-REM parasomnias involve confused behavior with or without vocalization and sleepwalking. Patients with REM sleep disorders often behave violently, dreaming that they are being attacked or chased, with the risk of injury during these episodes. Polysomnography is required to confirm the diagnosis.
**Transient global amnesia**	Usually occurs in people older than 50 years. Sudden onset of amnesia that lasts for several hours. Patients maintain alertness but are confused and ask questions repeatedly.
**Psychogenic non-epileptic seizures (PNES)**	PNES can be distinguished by spontaneous closing and opening of the eyes, associated with volitional head movements of „yes-yes” or „no-no” and prominent thrusting of the pelvis [28]. Correct diagnosis is usually based on the semiology of the event and the absence of an epileptiform EEG correlate.

## Data Availability

Not applicable.

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
