# Peer review of "State of the Art and Challenges in Epilepsy—A Narrative Review"

_jpm, 2023, doi:10.3390/jpm13040623_

Round 1

Reviewer 1 Report

The authors present a review about the healthcare in epilepsy..

The work work could be reconsidered for publication if major revision is carried out.

- In Table 2, I recommend detailing the meaning of the acronym as HIV at the end of it.

- In Table 2, when talking about immune etiologies, I would recommend the use of the term "autoimmune encephalitis" in a more general way, since there are other types of autoimmune encephalitis, in addition to those mentioned, that may cause epilepsy.

- In the "Diagnostic" section, it would be advisable to emphasize that the diagnosis is clinical and that complementary tests are support to it.

- In Table 4, in the MRI section, I recommend adding the use of a specific MRI protocol for epilepsy and I recommend changing what is written in "structural injuries", since there are many lesions that cause epilepsy, not only those located in the temporal lobe.

- In Table 4, there are other indications of lumbar puncture such as the suspicion of immune etiology that have not been detailed.

- In Table 5, by definition the TIAs are not associated with injury in neuroimaging, in that case it is is called cerebral infarction.

- In the "Complications and Prognosis" section, I would recommend adding information about cognitive decline in epilepsy.

Reviewer 2 Report

Review of Manuscript “The state of healthcare in epilepsy - A Narrative Review” submitted by Aida Mihaela Manole et al.

This was an interesting review which described in detail the etiology of epileptic seizures and the most common syndromes observed in children. There was information about different ways of epilepsy diagnostics and that many disorders can mimic epileptic seizures leading to various medical problems with a great impact on patients’ quality of life. Brief information was provided about the evolution, complication, prognosis of epilepsy, and its treatment.

My recommendations are listed below:

1.      As the review is focused on healthcare in epilepsy, I believe that this point of the manuscript is presented too shortly. It needs to be extended. More information can be added about different exercises and sports that can be done by epileptic patients. Different recommendations and problems can be outlined, as well as their effect on the quality of life.

2.      The title of the manuscript should be changed so, in this way to reveal more aspects of the manuscript, not only one.

3.      More information should be included about different comorbidities (such as cognitive dysfunction, depression, and anxiety) observed in patients with epilepsy and how health care helps to cope with them.

4.       Information about how healthcare helps these people to be integrated into society is also needed.

Round 2

Reviewer 1 Report

The authors have reviewed the manuscript and have made the corresponding corrections.

Reviewer 2 Report

Dear Authors,

All required changes have been done.